# Trivalent SARS-CoV-2 S1 Subunit Protein Vaccination Induces Broad Humoral Responses in BALB/c Mice

**DOI:** 10.3390/vaccines11020314

**Published:** 2023-01-31

**Authors:** Muhammad S. Khan, Eun Kim, Shaohua Huang, Thomas W. Kenniston, Andrea Gambotto

**Affiliations:** 1Department of Surgery, School of Medicine, University of Pittsburgh, Pittsburgh, PA 15213, USA; 2Department of Infectious Diseases and Microbiology, School of Public Health, University of Pittsburgh, Pittsburgh, PA 15261, USA; 3UPMC Hillman Cancer Center, Pittsburgh, PA 15232, USA; 4Department of Medicine, Division of Infectious Disease, School of Medicine, University of Pittsburgh, Pittsburgh, PA 15213, USA; 5Department of Microbiology and Molecular Genetics, School of Medicine, University of Pittsburgh, Pittsburgh, PA 15213, USA

**Keywords:** COVID-19, SARS-CoV-2, S1 recombinant protein, subunit vaccine, Omicron, Delta, next-generation COVID-19 vaccines, trivalent

## Abstract

This paper presents a novel approach for improving the efficacy of COVID-19 vaccines against emergent SARS-CoV-2 variants. We have evaluated the immunogenicity of unadjuvanted wild-type (WU S1-RS09cg) and variant-specific (Delta S1-RS09cg and OM S1-RS09cg) S1 subunit protein vaccines delivered either as a monovalent or a trivalent antigen in BALB/c mice. Our results show that a trivalent approach induced a broader humoral response with more coverage against antigenically distinct variants, especially when compared to monovalent Omicron-specific S1. This trivalent approach was also found to have increased or equivalent ACE2 binding inhibition, and increased S1 IgG endpoint titer at early timepoints, against SARS-CoV-2 spike variants when compared monovalent Wuhan, Delta, or Omicron S1. Our results demonstrate the utility of protein subunit vaccines against COVID-19 and provide insights into the impact of variant-specific COVID-19 vaccine approaches on the immune response in the current SARS-CoV-2 variant landscape. Particularly, our study provides insight into effects of further increasing valency of currently approved SARS-CoV-2 vaccines, a promising approach for improving protection to curtail emerging viral variants.

## 1. Introduction

The current COVID-19 pandemic, caused by severe acute respiratory syndrome coronavirus 2 (SARS-CoV-2), continues to have a significant impact on human and animal health globally [1,2,3]. The COVID-19 pandemic has over 637 million cases, 6.5 million deaths, with 12.9 billion COVID-19 vaccine doses administered across the human population, as of 29 November 2022 [4]. Approved COVID-19 vaccines have been a vital tool in reducing mortality and morbidity caused by SARS-CoV-2 infection. However, emerging immune-evasive SARS-CoV-2 variants, fueled by worldwide COVID-19 vaccine distribution inequalities, have left many low to middle income countries without access to variant-specific vaccines better suited for the evolving SARS-CoV-2 variant landscape [5,6,7,8]. Particularly, Delta (B.1.617.2), Omicron (BA.1), and Omicron sub lineages (BA.2, BA.4, BA.5, etc.) have shown to have the greatest resistance to vaccine-induced and infection-acquired immunity, leading to significant COVID-19 infection waves [8,9,10,11,12].

The spike (S) protein of SARS-CoV-2 has been the main target of currently approved COVID-19 vaccines and of most COVID-19 vaccines in development [13]. The S protein mediates virus binding and infection of susceptible cells through interaction with host receptor angiotensin-converting enzyme 2 (ACE2) [14]. The S protein is composed of two subunits, the S1 subunit which contains the receptor binding domain (RBD) that binds to ACE2, and the S2 subunit that allows for cell fusion and viral entry [15,16]. It has been well established that antibodies targeting the S protein, and the RBD within the S1 subunit, are able to block the binding of SARS-CoV-2 to the cell receptor and prevent infection of susceptible cells [17,18,19,20,21]. Protein subunit vaccine approaches against COVID-19 are highly favorable for worldwide equitable distribution due to their low cost per dose, relative thermostability, and excellent safety profile [22,23,24,25]. Our previously published reports on vaccines against SARS-CoV-1, Middle-East respiratory syndrome coronavirus (MERS-CoV), and SARS-CoV-2 have demonstrated the ability of S1 subunit targeting vaccines to generate neutralizing antibody responses against Beta coronaviruses [26,27,28,29,30].

Of particular interest is the investigation of novel COVID-19 vaccines, which may be able to induce broader antibody responses against multiple variants through multivalent vaccine immunization. A multivalent vaccine is a traditional approach used to increase antigen coverage against ever-changing pathogens such as COVID-19 [31,32,33]. However, it is necessary to investigate whether increasing valency of COVID-19 vaccines decreases the overall potency of the immune response or abrogates the per-variant host-antibody response. Indeed, a bivalent COVID-19 vaccine approach, through the mRNA platform, has been shown to have increased immunogenicity when compared to the monovalent approach in humans [34]. Trivalent vaccine approaches have been shown to increase immunogenicity of various vaccines, especially in the context of influenza [35,36,37,38,39]. Preclinical trivalent COVID-19 vaccines have also been shown to have increased immunogenicity when compared to monovalent approaches; however, these studies did not incorporate SARS-CoV-2 Omicron BA.1 variant of concern (VOC), an important piece of information explored by our study [37,38].

Here, we compared the immunogenicity of wild-type Wuhan spike S1 (WU-S1RS09cg), Delta variant-specific spike S1 (Delta S1-RS09cg), and Omicron variant-specific spike S1 (OM S1-RS09cg) subunit protein vaccines delivered either as a monovalent antigen or a combination of the three in trivalent antigen form (Wu/Delta/OM S1-RS09cg). We found that while monovalent vaccination resulted in substantial humoral response against S, a trivalent approach induced a broader humoral response with more coverage against antigenically distinct variants particularly in the context of monovalent Omicron-specific S1. The trivalent approach of Wu/Delta/OM S1-RS09cg showed increased ACE2 binding inhibition, and increased S1 IgG endpoint titer at early timepoints, against Wuhan and Delta S than monovalent OM S1-RS09cg. Our studies demonstrate the utility of protein subunit vaccines against COVID-19 and contribute insights into the impact of variant-specific COVID-19 vaccine approaches on the immune response in the context of the current SARS-CoV-2 variant landscape.

## 2. Materials and methods

### 2.1. Construction of Recombinant Protein Expressing Vectors

The coding sequence for SARS-CoV-2-S1 amino acids 1 to 661; having C-terminal tag known as ‘C-tag’, composed of the four amino acids (aa), glutamic acid-proline-glutamic acid-alanine (E-P-E-A) flanked with Sal I & Not I was codon-optimized using UpGene algorithm for optimal expression in mammalian cells [28,40]. The construct also contained a Kozak sequence (GCCACC) at the 5′ end. The plasmid, pAd/SARS-CoV-2-S1 was then created by subcloning the codon-optimized SARS-CoV-2-S1 inserts into the shuttle vector, pAdlox (GenBank U62024), at Sal I/Not I sites. The plasmid constructs were confirmed by DNA sequencing.

### 2.2. Transient Production in Expi293 Cells

pAd/S1RS09cg proteins were amplified and purified using ZymoPURE II plasmid maxiprep kit (Zymo Research). For Expi293 cell transfection, we used ExpiFectamieTM 293 Transfection Kit (ThermoFisher, Waltham, MA, USA) and followed the manufacturer’s instructions. Cells were seeded 3.0 × 10^6^ cells/mL one day before transfection and grown to 4.5–5.5 × 10^6^ cells/mL. 1 μg of DNA and ExpiFectamine mixtures per 1 mL culture were combined and incubated for 15 min before adding into 3.0 × 10^6^ cells/mL culture. At 20 h post-transfection, enhancer mixture was added, and culture was shifted to 32 °C. The supernatants were harvested 5 days post transfection and clarified by centrifugation to remove cells, filtration through 0.8 μm, 0.45 μm, and 0.22 μm filters and either subjected to further purification or stored at 4 °C before purification.

### 2.3. Purification of Recombinant Proteins 

The recombinant proteins were purified using a CaptureSelect^TM^ C-tagXL Affinity Matrix prepacked column (ThermoFisher) and followed the manufacturer’s guideline [41]. Briefly, The C-tagXL column was conditioned with 10 column volumes (CV) of equilibrate/wash buffer (20 mM Tris, pH 7.4) before sample application. Supernatant was adjusted to 20 mM Tris with 200 mM Tris (pH 7.4) before being loaded onto a 5-mL prepacked column per the manufacturer’s instructions at 5 mL/min rate. The column was then washed by alternating with 10 CV of equilibrate/wash buffer, 10 CV of strong wash buffer (20 mM Tris, 1 M NaCl, 0.05% Tween-20, pH 7.4), and 5 CV of equilibrate/wash buffer. The recombinant proteins were eluted from the column by using elution buffer (20 mM Tris, 2 M MgCl_2_, pH 7.4). The eluted solution was concentrated and desalted with preservative buffer (PBS) in an Amicon Ultra centrifugal filter devices with a 50,000 molecular weight cutoff (Millipore, Burlington, MA, USA). The concentrations of the purified recombinant proteins were determined by the Bradford assay using bovine serum albumin (BSA) as a protein standard, aliquoted, and stored at −80 °C until use.

### 2.4. SDS-PAGE, Silver Staining, and Western Blot 

The purified proteins were subjected to sodium dodecyl sulfate polyacrylamide gel electrophoresis (SDS-PAGE), Silver Staining, and Western blot. Briefly, after the supernatants were boiled in Laemmli sample buffer containing 2% SDS with beta- mercaptoethanol (β-ME), the proteins were separated by Tris-Glycine SDS-PAGE gels and transferred to nitrocellulose membrane. After blocking for 1 h at room temperature (RT) with 5% non-fat milk in TBS-T, rabbit anti-SARS-CoV spike polyclonal antibody (1:3000) (Sino Biological), or rabbit anti-SARS-CoV nucleoprotein (1:3000) (Sino Biological) was added and incubated overnight at 4 °C as primary antibody, and horseradish peroxidase (HRP)-conjugated goat anti-rabbit IgG (1:10,000) (Jackson immuno research) was added and incubated at RT for 1 hs as secondary antibody. After washing, the signals were visualized using ECL Western blot substrate reagents and iBright 1500 (Thermo Fisher).

### 2.5. Animals and Immunization

At week 0 female BALB/c mice (n = 5 animals per group) were bled from retro-orbital vein and primed with 45 μg of either WU S1-RS09cg, Delta S1-RS09cg, OM S1-RS09cg, or trivalent WU/Delta/OM S1-RS09cg. Mice were bled on week 3 and received a homologous booster of 45 μg. Mice were bled on week 5, 7, 9, 12, 16, and 20. Mice were maintained under specific pathogen-free conditions at the University of Pittsburgh, and all experiments were conducted in accordance with animal use guidelines and protocols approved by the University of Pittsburgh’s Institutional Animal Care and Use (IACUC) Committee.

### 2.6. ELISA

Sera from all mice were collected prior to immunization (week 0) and at weeks indicated after immunization and evaluated for SARS-CoV-2-S1-specific IgG, IgG1, and IgG2a antibodies using ELISA [28]. Briefly, ELISA plates were coated with 200 ng of recombinant SARS-CoV-2-S1 protein per well overnight at 4 °C in carbonate coating buffer (pH 9.5) and then blocked with PBS-T and 2% bovine serum albumin (BSA) for one hour. For ELISA coating antigens, Wuhan S1 was purchased from Sino Biological, Delta S1cg was produced by our lab, and Omicron S1-RS09cg was used to elucidate Omicron (BA.1) specific response. Mouse sera were serially diluted in PBS-T with 1% BSA and incubated overnight. After the plates were washed, anti-mouse IgG-horseradish peroxidase (HRP) (1:10,000, SantaCruz, Dallas, Texas, USA) was added to each well and incubated for 60 min. The plates were washed three times, developed with 3,3′5,5′-tetramethylbenzidine, and the reaction was stopped. Next, absorbance was determined at 450 nm using a plate reader. For IgG1 and IgG2a ELISAs, mouse sera were diluted in PBS-T with 1% BSA and incubated overnight. After the plates were washed, biotin-conjugated IgG1 and IgG2a (1:1000, eBioscience, San Diego, CA, USA) and biotin horseradish peroxidase (Av-HRP) (1:50,000, Vector Laboratories, Newark, CA, USA) were added to each well and incubated for 1 h. The plates were washed three times and developed with 3,3′5,5′-tetramethylbenzidine, the reaction was stopped, and absorbance at 450 nm was determined using a plate reader. ELISA data graphed is relative to preimmunization sera, using week 0 sera as the standardized cutoff.

### 2.7. ACE2 Blocking Assay

Antibodies blocking the binding of SARS-CoV-2 spike including Wuhan and spikes from immune evasive variants; BA.1, BA.2, AY.4 (Delta lineage), BA.3, BA.1 + R346K mutation, BA.1 + L52R mutation, B.1.1.7 (Alpha), B.1.351 (Beta), and B.1.1640.2 to ACE2 were detected with a V-PLEX SARS-CoV-2 Panel X (ACE2) Kit (Meso Scale Discovery (MSD)) according to the manufacturer’s instructions. The assay plate was blocked for 30 min and washed. Serum samples were diluted (1:25, 1:100 or 1:400) and 25 μL were transferred to each well. The plate was then incubated at room temperature for 60 min with shaking at 700 rpm, followed by the addition of SULFO-TAG conjugated ACE2, and continued incubation with shaking for 60 min. The plate was washed, 150 μL MSD GOLD Read Buffer B was added to each well, and the plate was read using the QuickPlex SQ 120 Imager. Electrochemiluminescent values (ECL) were generated for each sample. Results were calculated as % inhibition compared to the negative control for the ACE2 inhibition assay, and % inhibition is calculated as follows: % neutralization = 100 × (1 − (sample signal/negative control signal)).

### 2.8. Statistical Analysis

Statistical analyses were performed using GraphPad Prism v9 (San Diego, CA, USA). Antibody endpoint titers and neutralization data were analyzed by Kruskal-Wallis test, followed by Dunn’s multiple comparisons. Significant differences are indicated by * *p* < 0.05. Comparisons with non-significant differences are not indicated.

## 3. Results

### 3.1. Design and Expression of Recombinant Proteins

Recombinant proteins of SARS-CoV-2-S1, pAd/S1 Wu, pAd/S1 Delta, pAd/S1 Omicron (BA.1) were generated by subcloning the codon-optimized SARS-CoV-2-S1 gene having C-tag into the shuttle vector, pAd (GenBank U62024) at SalI and NotI sites (Figure 1A). To determine SARS-CoV-2-S1 expression and purity post C-tagXL affinity matrix purification, proteins were separated by 10% SDS-PAGE and assessed by silver staining (Figure 1B). The purified recombinant proteins WU S1-RS09cg (lane 1), Delta S1-RS09cg (lane 2), and Omicron S1-RS09cg (lane 3) were visualized at their expected glycosylated monomeric molecular weights of about 110 kDa under the denaturing reduced conditions. The proteins were also recognized by a polyclonal anti-spike SARS-CoV-2 antibody through western blot (Figure 1C).

### 3.2. Protein Subunit SARS-CoV-2 S1 Vaccines Induce Robust and Cross-Variant Binding IgG Responses

To assess the magnitude of the antibody response and the long-term persistence of immunogenicity, we first determined Wuhan, Delta, and Omicron (BA.1) specific IgG antibody endpoint titers (EPT) in the sera of vaccinated mice. Mice were prime and boosted on week 3 with either 45 μg of WU S1-RS09cg, Delta S1-RS09cg, OM S1-RS09cg, or a trivalent cocktail of the three antigens (15 μg WU S1-RS09cg, 15 μg Delta S1-RS09cg, 15 μg OM S1-RS09cg) in a single immunization. We collected serum samples from all mice prior to immunization, which were used set the endpoint titer cutoff for all antibody ELISA’s [42]. Serum samples collected on weeks 3, 5, 7, 9, 12, 16, and 20 after prime immunization were serially diluted to determine SARS-CoV-2-S1-specific IgG titers against Wuhan S1 (Figure 2), Delta S1 (Figure 3), and Omicron BA.1 (Figure 4) for each immunization group using ELISA.

Against Wuhan S1, all vaccinated groups had significantly higher geometric mean Wuhan S1 IgG EPT at week 5 when compared to week 3, illustrating the superior immunogenicity conferred by boost immunization (Figure 2, *p* < 0.05, Kruskal-Wallis test, followed by Dunn’s multiple comparisons). Interestingly, WU S1-RS09cg vaccinated mice achieved lower geometric mean Wuhan S1 IgG EPT by week 9 when compared to the other immunization groups (Figure 2). Indeed, trivalent WU/Delta/OM S1-RS09cg had increased Wuhan S1 IgG EPT when compared to monovalent OM S1-RS09cg at weeks 3, 5, and 7 (Figure 2). However, as waning of the immune response occurred, the trivalent WU/Delta/OM S1-RS09cg vaccinated mice reached similar geometric mean Wuhan S1 IgG EPT as monovalent OM S1-RS09cg vaccinated mice at week 9 with waning continuing to occur through week 20 (Figure 2).

Against Delta S1, Delta S1-RS09cg and Wu/Delta/OM S1-RS09cg vaccinated mice had the highest geometric mean Delta IgG EPT at week 3 (Figure 3). Only OM S1-RS09cg and WU/Delta/OM S1-RS09cg vaccinated mice achieved significantly higher geometric mean Delta S1 IgG EPT at week 5 when compared to week 3 (Figure 3, *p* < 0.05, Kruskal-Wallis test, followed by Dunn’s multiple comparisons). However, at week 5 OM S1-RS09cg vaccinated mice and Wu/Delta/OM S1-RS09cg vaccinated mice had the greatest geometric mean Delta IgG EPT (Figure 3). Interestingly, OM S1-RS09cg vaccinated mice and Wu/Delta/OM S1-RS09cg vaccinated mice Delta S1 IgG antibody response waned less from week 5 through week 20 than the other immunization groups (Figure 3).

Against Omicron S1, Delta S1-RS09cg, OM S1-RS09cg, and Wu/Delta/OM S1-RS09cg vaccinated mice had significantly increased Omicron S1 IgG EPT at week 5 when compared to week 3 (Figure 4, *p* < 0.05, Kruskal-Wallis test, followed by Dunn’s multiple comparisons). Both OM S1-RS09cg and trivalent Wu/Delta/OM S1-RS09cg vaccinated mice achieved the highest geometric mean Omicron S1 IgG EPT by week 3 and through week 20 (Figure 4). A difference between OM-S1RS09cg and Wu/Delta/OM S1-RS09cg geometric mean Omicron S1 IgG EPT occurred at week 9, with OM-S1RS09cg vaccinated mice having modestly higher EPT than trivalent vaccinated mice through week 20 (Figure 4).

To assess whether the IgG antibody response was Th1- or Th2-specific, serum samples were collected at week 5 and serially diluted to determine Wuhan and BA.1-specific S1, IgG1 (indicating a Th2 bias) and IgG2a (indicating a Th1 bias) endpoint titers for each immunization group (Figure 5A–D). Interestingly, against Wuhan S1 all vaccinated mice groups achieved similar IgG1 and IgG2a geometric mean S1 IgG1 and IgG2a EPT, with no significant differences between groups (Figure 5A,B). Differences between vaccine groups were illuminated against BA.1 S1 (Figure 5C,D). Both OM-S1RS09cg and trivalent WU/Delta/OM S1-RS09cg vaccinated mice had the greatest geometric mean Omicron-S1 IgG1 EPT than WU S1-RS09cg and Delta S1-RS09cg vaccinated mice (Figure 5C,D). OM S1-RS09cg vaccinated mice achieved the highest BA.1 S1 IgG2a geometric mean EPT (Figure 5C,D). As expected for unadjuvanted protein subunit vaccine in BALB/c mouse, all vaccinated groups had a trend to a IgG1 dominant IgG response, indicating a Th2 bias.

These results suggest that Wuhan S1-RS09cg, Delta S1-RS09cg, OM-S1RS09cg, and trivalent WU/Delta/OM S1-RS09cg all stimulated a robust IgG binding antibody response in BALB/c mice against Wuhan S1, Delta S1, and Omicron (BA.1) S1.

### 3.3. ACE2 Binding Inhibition 

Competitive immunoassays for quantifying inhibition of the spike-ACE2 interaction have been shown to correlate well with live-virus neutralizing tests and serve as a convenient multiplex method to determine the neutralizing capacity of vaccinated sera [43,44,45,46]. To investigate the neutralizing capabilities of antibodies induced by vaccination we used the Meso Scale Discovery (MSD) V-PLEX SARS-CoV-2 (ACE2) Kit. This measures the inhibition of binding between angiotensin converting enzyme-2 (ACE2) and trimeric spike protein of SARS CoV-2 variants. We used kit Panel 25 including Wuhan S and spikes from immune evasive variants; BA.1, BA.2, AY.4 (Delta lineage), BA.3, BA.1 + R346K mutation, BA.1 + L52R mutation, B.1.1.7 (Alpha), B.1.351 (Beta), and B.1.1640.2. Sera from vaccinated animals were examined at week 5 and week 7, the peak of the IgG antibody responses (Figure 2, Figure 3 and Figure 4). Figure 6A,B depict the median ACE2-binding percent inhibition of each vaccinated mice group sera at week 5 and week 7, respectively. Figure 6C–F depict each vaccination group ACE2-binding percent inhibition individually; WU-S1RS09cg vaccinated mice (Figure 6C), Delta S1-RS09cg vaccinated mice (Figure 6D), OM-S1RS09cg vaccinated mice (Figure 6E), and trivalent WU/Delta/OM S1-RS09cg vaccinated mice (Figure 6E). Antibodies blocking ACE2 and trimeric S binding were detected in all vaccination groups. Interestingly, WU S1-RS09cg vaccinated mice achieved the lowest median ACE2-binding inhibition against Wuhan S, AY.4 (Delta), B.1.1.7, B.1.351, and B.1.640.2 at weeks 5 and 7 when compared to other vaccination groups (Figure 6A,B). Delta S1-RS09cg vaccinated mice had a robust ACE2-binding inhibition response against WU S, AY.4 (Delta), B.1.1.7, B.1.351, and B.1.640.2; with a diminished response against Omicron (BA.1) and Omicron sub lineages (Figure 6A,B). OM-S1RS09cg vaccinated mice had moderate to high median ACE2 binding inhibition against all S tested, with robust inhibition of ACE2 binding of Omicron and Omicron sub lineages, when compared to other vaccination groups at weeks 5 and 7 (Figure 6A,B). Trivalent WU/Delta/OM S1-RS09cg, when compared to the monovalent counterparts, had increased coverage of median ACE-2 binding inhibition against all variants tested (Figure 6A,B). Notably, when comparing OM S1-RS09cg vaccinated mice to trivalent WU/Delta/OM S1-RS09cg vaccinated mice, trivalent vaccinated mice had greater median ACE2-binding percent inhibition against WU S, BA.2, AY.4 (Delta), BA.1 + R346K, B.1.1.7, B.1.351, and B.1.640.2 (Figure 6A,B).

To combine the data on Wuhan, Delta, and Omicron (BA.1) S binding IgG EPT and ACE2-percent binding inhibition we plotted the respective mean values at week 5 against each other (Figure 7A–C). In the context of Wuhan S, Delta S1-RS09cg and trivalent WU/Delta/OM S1-RS09cg vaccinated mice grouped together with the highest mean S1 IgG EPT and mean ACE2 binding inhibition (Figure 7A). Against Delta S, Delta S1-RS09cg and trivalent WU/Delta/OM S1-RS09cg vaccinated mice group together with the highest mean S1 IgG EPT and mean ACE2 binding inhibition (Figure 7B). For BA.1 S, OM S1-RS09cg and trivalent WU/Delta/OM S1-RS09cg vaccinated mice grouped together with the highest mean S1 IgG EPT and mean ACE2 binding inhibition (Figure 7C). 

Taken together, a prime and boost of non-adjuvanted recombinant S1 protein subunit vaccine induced a robust humoral antibody response against SARS-CoV-2 in BALB/c mice. Particularly, trivalent WU/Delta/OM vaccinated mice induced a broad and cross-reactive neutralizing antibodies against SARS-CoV-2 variants with increased breadth when compared to monovalent WU S1-RS09cg, Delta S1-RS09cg, and OM S1-RS09cg vaccinated mice.

## 4. Discussion

As SARS-CoV-2 variants continue to emerge more vaccination platforms against SARS-CoV-2, which induce a broader immune response covering multiple variants, will be necessary [8,9,11,12,43,47]. Further, as COVID-19 booster doses are distributed, it will be critical to ensure that global vaccine equity is met [48,49]. Protein subunit vaccines are ideal for worldwide distribution due to their excellent safety, low cost, scalability, and thermostability [23,24,50]. Protein subunit vaccine platforms can be further improved through use of alternative vaccine delivery methods such as intranasal or intradermal vaccination, with microneedle arrays [28,51]. The versatility of protein subunit vaccines lends to their utility for mass distribution and vaccination.

In this study, we demonstrate the robust antibody response elicited by our unadjuvanted S1 protein subunit vaccine in BALB/c mice. Wuhan S1-RS09cg, Delta S1-RS09cg, OM-S1RS09cg, and trivalent WU/Delta/OM S1-RS09cg vaccinated mice all elicited a robust IgG binding antibody response against Wuhan S1, Delta S1, and Omicron (BA.1) S1. Particularly, trivalent WU/Delta/OM S1-RS09cg vaccinated mice mounted cross-reactive ACE2 binding inhibiting antibodies against SARS-CoV-2 variants with increased breadth when compared to monovalent WU S1-RS09cg, Delta S1-RS09cg, and OM S1-RS09cg vaccinated mice. We believe that this gives credence to investigating SARS-CoV-2 vaccines that are multivalent to expand variant specific immune responses. Our data also suggests that increasing valency of SARS-CoV-2 vaccines may not reduce magnitude of the individual variant immune response, a key added piece of information for development of next-generation SARS-CoV-2 vaccines. A particularly unexpected result of our study is the low immunogenicity of our WU S1-RS09cg vaccine against Wuhan S1, and other VOCs, when compared to Delta and OM S1-RS09cg. Indeed, Delta and Omicron (BA.1) mutations in S have been shown to increase pathogenicity and S fusogenicity, along with increased ACE2 binding to S, when compared to wild-type Wuhan SARS-CoV-2 [47,52,53,54,55]. We hypothesize that this increased ACE2 binding by Delta and Omicron S may explain the increased immunogenicity exhibited by Delta and OM S1-RS09cg when compared to Wuhan S1-RS09cg, however, this will need to be explored further. Furthermore, Omicron BA.1 spike G446S mutation has been shown to potentiate antiviral T-cell recognition which may further explain the increased immunogenicity demonstrated by our OM S1-RS09cg and trivalent vaccine candidates [56].

The IgG isotype of the induced IgG antibodies skew to be IgG1 dominant, indicating a Th2-type bias. Indeed, BALB/c mice are the prototypical Th2-type mouse strain which necessitates the investigation of this protein subunit vaccine in additional animal models to examine the risk of vaccine-associate enhanced respiratory disease (VAERD) [57]. Our previous research has suggested that a booster of unadjuvanted subunit vaccine after an Adenoviral prime vaccine might avoid Th2-based immune response and the occurrence of VAERD [30]. Indeed, there have been numerous Adenoviral vector vaccine platforms used in the SARS-CoV-2 pandemic and constitute a large population necessitating variant-specific boosting [7,58]. Further so, the Th1- and Th2-type immune response may be further augmented using an adjuvant. In the context of SARS-CoV-2 vaccines there have been numerous adjuvants that have shown beneficial effects on immunogenicity [59,60,61,62]. Interestingly, a AS01-like adjuvanted SARS-CoV-2 subunit vaccine enhanced Th1-type IgG2a isotype and IFN-γ secreting T cell immune responses in BALB/c mice when compared to unadjuvanted control [63].

An important limitation regarding our study is the lack of T-cell immunity investigation and SARS-CoV-2 challenge, which were not performed to assess the protection ability of our vaccine constructs. S-specific binding antibodies were positively correlated with S-specific T-cell responses indicating induction of T cell immune response by our vaccine constructs [64]. We chose to focus on the induction of antibodies because they are the hypothesized correlate of protection against severe COVID-19 [20]. Furthermore, prior studies have shown the positive correlation and high concordance between binding antibodies and traditional virus-based microneutralization tests [30]. Our past work has also shown the positive correlation between the MSD ACE2 binding inhibition and virus-based microneutralization tests [30]. As a conventional and multiplex test, measurement of competitive immunoassay for quantifying inhibition of the spike-ACE2 interaction can serve as a surrogate for traditional virus-based microneutralization tests with high levels of correlation [43,45,46]. Our future studies will probe the protection ability elicited by our monovalent and trivalent vaccines through challenge studies using BALB/c mice and K18-hACE2 mice. The BALB/c mouse model of SARS-CoV-2 infection only supports infection of SARS-CoV-2 variants that carry the N501Y variant, necessitating the use of hACE2-transgenic mice to evaluate protection efficiency against other variants [65].

Overall, this study illustrates the potential of subunit protein vaccine targeting SARS-CoV-2-S1 as it induces significant induction of humoral immune responses against SARS-CoV-2 even without adjuvant. Particularly, immunizing with trivalent WU/Delta/OM S1-RS09cg increased binding antibodies and ACE2-binding inhibiting antibodies against SARS-CoV-2 variant spikes versus monovalent approaches. Furthermore, combining our protein subunit protein vaccine targeting SARS-CoV-2-S1 with an immunostimulatory adjuvant should provide even higher levels of immunogenicity when compared to the unadjuvanted studies presented here. Our findings support the use of trivalent Wuhan, Delta, and Omicron targeting COVID-19 vaccines to increase variant antigenic coverage.

## Figures and Tables

**Figure 1 vaccines-11-00314-f001:**
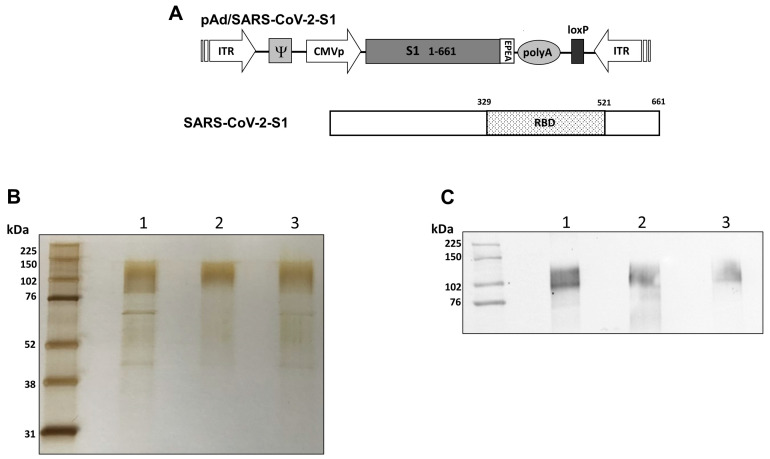
Construction of recombinant SARS-CoV-2-S1 protein expressing plasmid. (**A**) Shuttle vector carrying the codon-optimized wild-type (Wuhan), Delta variant, and Omicron variant (BA.1) SARS-CoV-2-S1 gene encoding N-terminal 1-661 with c-tag (EPEA) was designed as shown in the diagram. ITR: Inverted terminal repeat; RBD: receptor binding domain. (**B**) Silver-stained reducing SDS-PAGE gel of purified Expi293 cell derived Wuhan (WU) S1-RS09cg (Lane 1), Delta S1-RS09cg (Lane 2), and Omicron (OM) S1-RS09cg (Lane 3). (**C**) Detection of the SARS-CoV-2-S1 proteins by western blot with purified proteins using anti S SARS-CoV-2 polyclonal antibody; Wuhan (WU) S1-RS09cg (Lane 1), Delta S1-RS09cg (Lane 2), and Omicron (OM) S1-RS09cg (Lane 3).

**Figure 2 vaccines-11-00314-f002:**
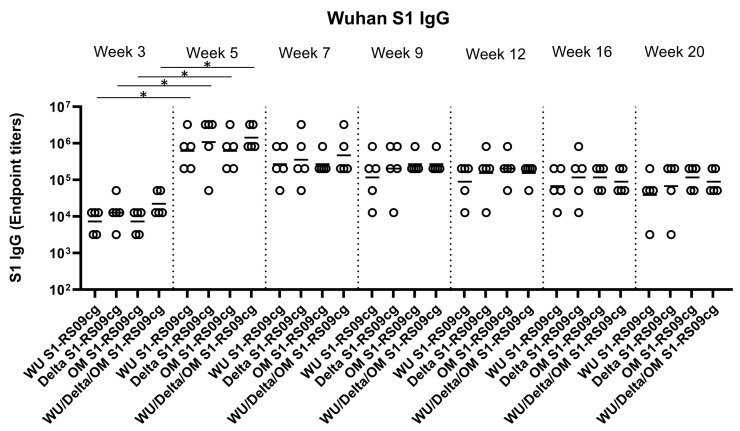
Wuhan S1-specific IgG antibody responses in mice after prime-boost immunization in BALB/c mice. BALB/c mice (n = 5 mice per groups) were immunized intramuscularly with 45 μg of either WU S1-RS09cg, Delta S1-RS09cg, OM S1-RS09cg or trivalent WU/Delta/OM S1-RS09cg and received a homologous booster at week 3. On weeks 3, 5, 7, 9, 12, 16, and 20 sera from mice were collected, serially diluted (200×), and tested for the presence of Wuhan SARS-CoV-2-S1-specific IgG antibody levels by ELISA. Significance was determined by Kruskal-Wallis test followed by Dunn’s multiple comparisons (* *p* < 0.05). Horizontal solid lines represent geometric mean antibody titers. Serum collected on week 0, prior to immunization, were used to set the ELISA endpoint titer cutoff.

**Figure 3 vaccines-11-00314-f003:**
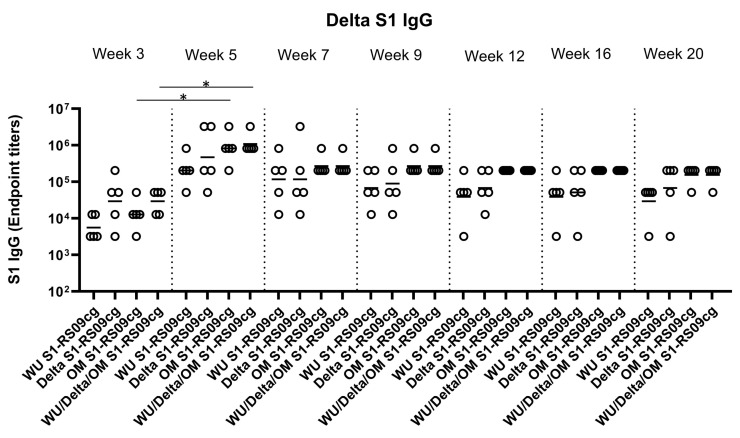
Delta S1-specific IgG antibody responses in mice after prime-boost immunization in BALB/c mice. BALB/c mice (n = 5 mice per groups) were immunized intramuscularly with 45 μg of either WU S1-RS09cg, Delta S1-RS09cg, OM S1-RS09cg or trivalent WU/Delta/OM S1-RS09cg and received a homologous booster at week 3. On weeks 3, 5, 7, 9, 12, 16, and 20 sera from mice were collected, serially diluted (200×), and tested for the presence of Delta variant SARS-CoV-2-S1-specific IgG antibody levels by ELISA. Significance was determined by Kruskal-Wallis test followed by Dunn’s multiple comparisons (* *p* < 0.05). Horizontal solid lines represent geometric mean antibody titers. Serum collected on week 0, prior to immunization, were used to set the ELISA endpoint titer cutoff.

**Figure 4 vaccines-11-00314-f004:**
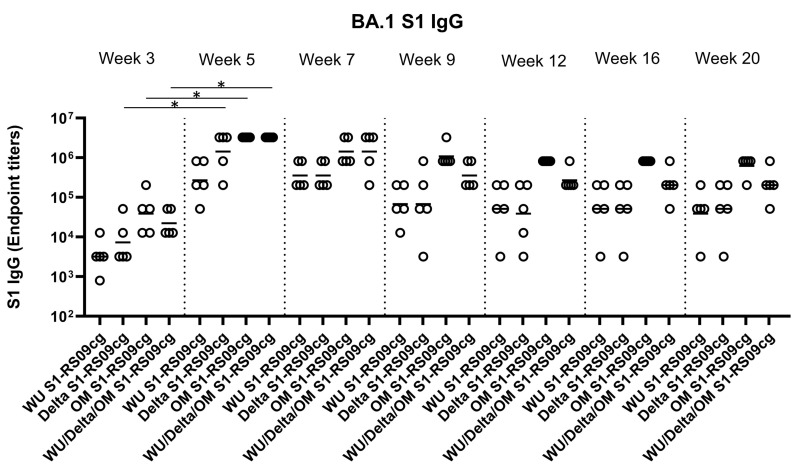
Omicron (BA.1) S1-specific IgG antibody responses in mice after prime-boost immunization in BALB/c mice. BALB/c mice (n = 5 mice per groups) were immunized intramuscularly with 45 μg of either WU S1-RS09cg, Delta S1-RS09cg, OM S1-RS09cg or trivalent WU/Delta/OM S1-RS09cg and received a homologous booster at week 3. On weeks 3, 5, 7, 9, 12, 16, and 20 sera from mice were collected, serially diluted (200×), and tested for the presence of Omicron (BA.1) SARS-CoV-2-S1-specific IgG antibody levels by ELISA. Significance was determined by Kruskal-Wallis test followed by Dunn’s multiple comparisons (* *p* < 0.05). Horizontal solid lines represent geometric mean antibody titers. Serum collected on week 0, prior to immunization, were used to set the ELISA endpoint titer cutoff.

**Figure 5 vaccines-11-00314-f005:**
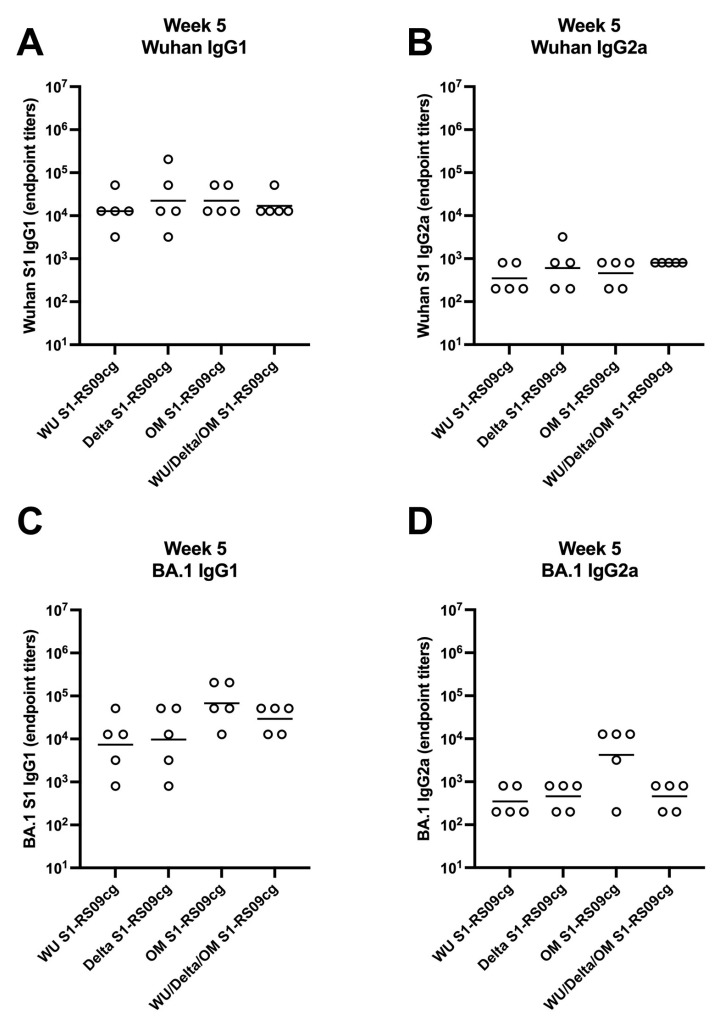
Wuhan and Omicron (BA.1)-specific IgG1 and IgG2a antibody responses in mice after prime-boost immunization in BALB/c mice. BALB/c mice (n = 5 mice per groups) were immunized intramuscularly with 45 μg of either WU S1-RS09cg, Delta S1-RS09cg, OM S1-RS09cg or trivalent WU/Delta/OM S1-RS09cg and received a homologous booster at week 3. On weeks 3, 5, 7, 9, 12, 16, and 20 sera from mice were collected, serially diluted (200×), and tested for the presence of Wuhan and Omicron (BA.1) SARS-CoV-2-S1-specific IgG1 and IgG2a antibody levels by ELISA. Significance was determined by Kruskal-Wallis test followed by Dunn’s multiple comparisons (*p* < 0.05). Horizontal solid lines represent geometric mean antibody titers. Serum collected on week 0, prior to immunization, were used to set the ELISA endpoint titer cutoff. (**A**) Week 5 Wuhan S1 IgG1; (**B**) Week 5 Wuhan S1 IgG2a; (**C**) Week 5 BA,1 S1 IgG1; (**D**) Week 5 BA.1 S1 IgG2a.

**Figure 6 vaccines-11-00314-f006:**
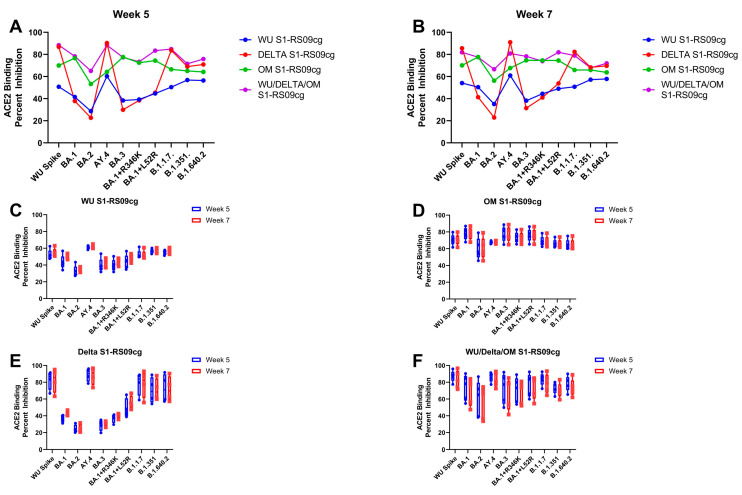
Percent ACE2 binding inhibition of neutralizing antibodies against SARS-CoV-2 variant elicited by monovalent and trivalent immunization in BALB/c mice. Antibodies in sera capable of neutralizing the interaction between SARS-CoV-2 Wuhan, BA.1, BA.2, AY.4 (Delta lineage), BA.3, BA.1 + R346K mutation, BA.1 + L52R mutation, B.1.1.7 (Alpha), B.1.351 (Beta), and B.1.1640.2. variant spike and ACE2 were examined in all animals at week 5 and week 7. (**A**) Per immunization group median ACE2 binding percent inhibition of WU S1-RS09cg vaccinated mice (blue dots), Delta S1-RS09cg vaccinated mice (red dots), OM S1-RS09cg vaccinated mice (green dots), and OM S1-RS09cg vaccinated mice (purple dots) at week 5 against each SARS-CoV-2 variant. (**B**) Per immunization group median ACE2 binding percent inhibition of WU S1-RS09cg vaccinated mice (blue dots), Delta S1-RS09cg vaccinated mice (red dots), OM S1-RS09cg vaccinated mice (green dots), and OM S1-RS09cg vaccinated mice (purple dots) at week 7 against each SARS-CoV-2 variant. Figure 6C–F depict each vaccination group individual mice ACE2-binding percent inhibition against all variants at week 5 (blue box and whisker plot) and week 7 (red box and whisker plot). Figure 6C WU S1-RS09cg elicited antibodies percent ACE2 binding inhibition. Figure 6D Delta S1-RS09cg elicited antibodies percent ACE2 binding inhibition. Figure 6E OM S1-RS09cg elicited antibodies percent ACE2 binding inhibition. Figure 6F WU/Delta/OM S1-RS09cg elicited antibodies percent ACE2 binding inhibition. Box and whisker plots represent the median and upper and lower quartile (box) with min and max (whiskers).

**Figure 7 vaccines-11-00314-f007:**
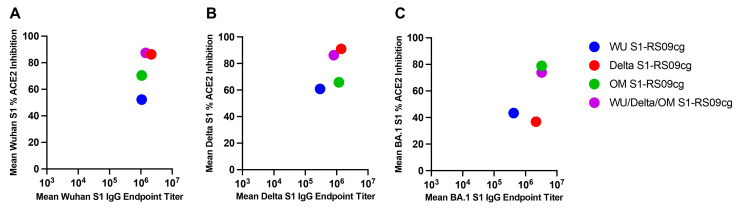
Variant-specific binding IgG and ACE2-percent binding inhibition in BALB/c mice. Wuhan, Delta, and Omicron (BA.1) S binding IgG mean EPT and mean ACE2-percent binding inhibition per variant plotted against the respective mean values at week 5. (**A**) Wuhan Week 5 binding IgG and ACE2-percent binding inhibition; (**B**) Delta Week 5 binding IgG and ACE2-percent binding inhibition; (**C**) BA.1 Week 5 binding IgG and ACE2 percent binding inhibition.

## Data Availability

The data that support the finding of this study are available from the corresponding author upon reasonable request.

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
