# Peer review of "Trivalent SARS-CoV-2 S1 Subunit Protein Vaccination Induces Broad Humoral Responses in BALB/c Mice"

_vaccines, 2023, doi:10.3390/vaccines11020314_

Round 1

Reviewer 1 Report

The manuscript herein presented aims to disclose the potential immune benefits of the use of a trivalent SARS-CoV-2 S1 vaccine. The work is well structured and the results are in accordance with the conclusions described by the authors. Nevertheless, I still have some comments and suggestions to address:

1. Statistics

Throughout the text and on the figures legends the authors describe the statistical methods used. However, the significant differences are never disclosed in the figures and its description in the text is difficult to follow. Please modify the figures with the statistical differences displayed and rearrange the text accordingly.

2. Figure 6 A and B

The authors have chosen a XY graphic representation for the display of the neutralization studies. The use of different antigens in the X axis is not appropriate for this graphic typology as the line between the dots have no meaning at all. Please, substitute this graphics representations with a more suitable typology (graphic or table or both)

3. unadjuvanted vs adjuvanted

In the last paragraph of the discussion (line 269), the authors state the potential of unadjuvanted trivalent subunit vaccine against SARS-CoV-2. In my opinion, this claim is correct in the context of the work herein presented but is misleading because, as the authors also discuss, adjuvanted vaccines will always provide more robust immune responses. The authors should moderate this claim and if possible, discuss the possibility of adjuvanted versions of  this trivalent vaccine provide even more positive results when compared to monovalent vaccines.

Author Response

1. Statistics

Throughout the text and on the figures legends the authors describe the statistical methods used. However, the significant differences are never disclosed in the figures and its description in the text is difficult to follow. Please modify the figures with the statistical differences displayed and rearrange the text accordingly.

We thank the reviewer for this suggestion. The figures have been updated to illustrate all significant differences.

2. Figure 6 A and B

The authors have chosen a XY graphic representation for the display of the neutralization studies. The use of different antigens in the X axis is not appropriate for this graphic typology as the line between the dots have no meaning at all. Please, substitute this graphics representations with a more suitable typology (graphic or table or both)

We thank the reviewer for this suggestion on Figure 6A and Figure 6B. We believe that this sort of graphical representation for the neutralization studies is an appropriate representation of the data. Figure 6A and Figure 6B aim to summarize the data illustrated through Figures 6C-6F which illustrate the specific response per vaccine construct. We believe the line between the dots serves two purposes; 1.) to illustrate the average effect of spike variants on percentage of neutralization per vaccine group - ie: for Delta S1-RS09cg there was an approximate 50% reduction in neutralization for BA.1 spike when compared to Wuhan spike and 2) aid the reader in following the average data for each vaccine group to not confuse the different vaccine groups summarized. For these reasons, we would like to keep Figure 6 as submitted.

3. unadjuvanted vs adjuvanted

In the last paragraph of the discussion (line 269), the authors state the potential of unadjuvanted trivalent subunit vaccine against SARS-CoV-2. In my opinion, this claim is correct in the context of the work herein presented but is misleading because, as the authors also discuss, adjuvanted vaccines will always provide more robust immune responses. The authors should moderate this claim and if possible, discuss the possibility of adjuvanted versions of  this trivalent vaccine provide even more positive results when compared to monovalent vaccines.

We thank the reviewer for this point of clarification. The manuscript discussion has been updated to clarify our claims of our trivalent subunit vaccine and have also included a sentence to highlight the ability of adjuvant to further drive the immunogenicity of our SARS-CoV-2-S1 protein vaccines.

Reviewer 2 Report

The Article should be published

The article describes the production of immunogens based on the S1 protein fragment of the surface glycoprotein SARS-CoV-2. In the work, the authors used immunogens both on the basis of single proteins (S1 subunit protein) and mixtures of all three. The resulting immunogens during immunization of mice caused the production of specific antibodies, while immunization with a mixture of immunogens produced antibodies reactive against the Wuhan, Delta, or Omicron variants.

The introduction is well documented.

Materials and Methods are described with enough details to allow others to replicate and build on published results.

The purpose of the study was achieved

Comments:

The article is not formatted according to the standards of the journal.

Author Response

The Article should be published

The article describes the production of immunogens based on the S1 protein fragment of the surface glycoprotein SARS-CoV-2. In the work, the authors used immunogens both on the basis of single proteins (S1 subunit protein) and mixtures of all three. The resulting immunogens during immunization of mice caused the production of specific antibodies, while immunization with a mixture of immunogens produced antibodies reactive against the Wuhan, Delta, or Omicron variants.

The introduction is well documented.

Materials and Methods are described with enough details to allow others to replicate and build on published results.

The purpose of the study was achieved

Comments:

The article is not formatted according to the standards of the journal.

We thank the reviewer for their synopsis of our work and recommendation for publishing. We have corrected the format of the manuscript to be in accordance of the journal.

Round 2

Reviewer 1 Report

The authors have answered all my queries and thus I recommend the manuscript publication at Vaccines.